# Type I Interferons as Joint Regulators of Tumor Growth and Obesity

**DOI:** 10.3390/cancers13020196

**Published:** 2021-01-07

**Authors:** Sandra Gessani, Filippo Belardelli

**Affiliations:** 1Center for Gender-Specific Medicine, Istituto Superiore di Sanità, 00161 Rome, Italy; 2Institute of Translational Pharmacology, CNR, 00133 Rome, Italy

**Keywords:** type I interferons, obesity, cancer, inflammation, tumor microenvironment, immunoregulation

## Abstract

**Simple Summary:**

The escalating global epidemic of overweight and obesity is a major public health and economic problem, as excess body weight represents a significant risk factor for several chronic diseases including cancer. Despite the strong scientific evidence for a link between obesity and cancer, the mechanisms involved in this interplay have not yet been fully understood. The aim of this review is to evaluate the role of type I interferons, a family of antiviral cytokines with key roles in the regulation of both obesity and cancer, highlighting how the dysregulation of the interferon system can differently affect these pathological conditions.

**Abstract:**

Type I interferons (IFN-I) are antiviral cytokines endowed with multiple biological actions, including antitumor activity. Studies in mouse models and cancer patients support the concept that endogenous IFN-I play important roles in the control of tumor development and growth as well as in response to several chemotherapy/radiotherapy treatments. While IFN-I signatures in the tumor microenvironment are often considered as biomarkers for a good prognostic response to antitumor therapies, prolonged IFN-I signaling can lead to immune dysfunction, thereby promoting pathogen or tumor persistence, thus revealing the “Janus face” of these cytokines in cancer control, likely depending on timing, tissue microenvironment and cumulative levels of IFN-I signals. Likewise, IFN-I exhibit different and even opposite effects on obesity, a pathologic condition linked to cancer development and growth. As an example, evidence obtained in mouse models shows that localized expression of IFN-I in the adipose tissue results in inhibition of diet–induced obesity, while hyper-production of these cytokines by specialized cells such as plasmacytoid dendritic cells in the same tissue, can induce systemic inflammatory responses leading to obesity. Further studies in mouse models and humans should reveal the mechanisms by which IFN-I can regulate both tumor growth and obesity and to understand the role of factors such as genetic background, diet and microbioma in shaping the production and action of these cytokines under physiological and pathological conditions.

## 1. Introduction

Interferon (IFN) was first identified more than 60 years ago as a factor released by virus-infected cells capable of inhibiting viral replication in target cells. Named for their capacity to “interfere” with viral replication, it is now well recognized that IFN also have distinct roles outside of infection and are involved in many biological processes [1]. Subsequently, several molecules belonging to the IFN family, released upon encounter with pathogens or various stimuli, have been identified and classified into different subtypes: type I IFN including α, β, ε, κ, and ω subtypes that bind IFNα/β receptor 1 (IFNAR1) and IFNAR2 subunits, type II IFN or IFN-γ that binds IFN-γ receptor 1 (IFNGR1), and type III IFN or IFN-λ, binding to IFN-λ receptor 1 and IL-10 receptor subunit β. Although type I IFN (IFN-I) are a major line of host defense against viruses and other pathogens, it is now clear that they can also drive context-specific responses to infection, which may be either beneficial or detrimental to the host [2,3,4]. Furthermore, dysregulation of the IFN-I system can elicit autoimmune diseases [5], and some evidence implicates IFN-I-dependent signaling as a key inflammatory driver in non-autoimmune diseases such as certain solid tumors and myocardial infarction [6,7].

IFN-I, including IFN-α and IFN-β, are the cytokines with the longest record of clinical use in patients with cancer and some viral and autoimmune diseases [8,9]. Almost all cells in the body can produce IFN-I following the recognition of molecules, such as foreign and self-nucleic acids and a minority of other non-nucleic-acids [collectively known as pathogen-associated molecular patterns (PAMP)], by the pathogen recognition receptors. Likewise, signals known as damage-associated molecular pathways (DAMP) are also known to stimulate IFN-I expression. Along with PAMP and DAMP, IFN-I can also be produced in response to rare physiologic stimuli such as colony stimulating factor (CSF)1, receptor activator of NF-kB ligand (RANK) and estrogens [10]. Expression of IFN-I is also dependent on the nutrient sensor mammalian target of rapamycin (mTOR) network as well as direct reprograming of lipid metabolic pathways [11]. On the other hand, IFN-I are potent modulators of cellular metabolism and biosynthetic reactions, highlighting a mutual relationship between IFN-I production and metabolic core processes [12,13]. Besides their induced expression, IFN-I are also constitutively produced at extremely low quantities and yet exert profound effects, mediated in part through modulation of signaling intermediates required for responses to diverse cytokines [14]. Worth of note, a basal systemic IFN response is maintained under physiological conditions through signals generated by the commensal microbiota to calibrate innate immune responses and maintain homeostasis [15,16,17]. As many cytokines, IFN-I induce balanced responses in which activating signals that induce antiviral states and promote immune responses are counterbalanced by suppressive signals that limit toxicity to the host. These balanced responses are fine-tuned by host factors at multiple levels and loss of this fine-tuning can result in sustained IFN signaling, immunosuppression and tissue damage, which has been implicated in pathogenesis of chronic viral infections, autoimmune diseases and cancer [18]. Noteworthy, the efficacy of several therapeutic strategies against cancer depends on the production and/or action of endogenous IFN-I [6,19].

Interestingly, dysfunctions of the IFN-I system have been observed in obesity, a pathological condition linked to cancer development and growth [20,21] and are associated with an increased susceptibility to infectious diseases and reduced efficacy of vaccine-induced immune responses. Overweight and obesity are significant risk factors for a number of chronic disorders, including cardiovascular diseases and type 2 diabetes mellitus (T2D) and, more importantly, several cancer types. This condition affects all stages of cancer development and may have a negative impact on the response to therapy. In this scenario, it is worth noting that IFN-I are emerging as novel players in the antitumor cascade to some solid cancers such as breast cancer, melanoma and colorectal cancer (CRC) with impairment of their expression/signaling pathway associated with disease outcome and decreased survival. Although a direct correlation between obesity-associated IFN-I system dysfunction and cancer development has not yet been reported, the immune alterations observed in obesity and the most recent unraveled role of IFN-I signaling impairment in the cancer cascade suggest a link between the immune regulatory capacity of these cytokines and both obesity and cancer.

In this review article, we will discuss the role of endogenous IFN-I in the control of both tumor growth and obesity highlighting how the dysregulation of the IFN-I system may result in different and even opposite effects in these pathological conditions.

## 2. Type I Interferons and Cancer

The first strong evidence of the antitumor effect of IFN-I in mouse tumor models was provided by Gresser and co-workers more than 50 years ago, by using partially purified preparations of IFN-α/β [22]. These findings and subsequent studies in mouse models led to many clinical studies in patients, firstly based on the use of natural IFN-α and subsequently of recombinant IFN-α subtypes [8]. For many years, the remarkable antitumor effects of IFN-α observed in patients with some hematological malignancies (especially hairy cell leukemia and chronic myeloid leukemia) as well as with certain solid tumors, including melanoma and renal cancer, contributed in maintaining a high interest of the scientific community as well as of patients and media on the clinical use of this cytokine in cancer. Today, the clinical use of IFN-α has been mostly replaced by new drugs and targeted therapies. Nevertheless, the most recent progress on cancer and IFN research keeps revealing new rationales and modalities of using IFN in cancer [23].

The pleiotropic nature of the multiple biological activities exerted by IFN-I on both tumor and host cells renders it difficult to fully understand which of the many effects and mechanisms observed under in vitro experimental conditions are really important in vivo in mediating the antitumor response in patients. IFN-I can exert antitumor activity by two main types of mechanisms: (i) acting on tumor cells, thus affecting their phenotype and growth properties (direct effects), and (ii) activating host cells, such as endothelial cells and cells of the innate and adaptive immunity, including dendritic cells (DC), natural killer (NK) and T cells, thus promoting an antitumor immune response (indirect effects).

By binding to IFN-I receptors on tumor cells, IFN-I can modulate gene expression by multiple mechanisms, inducing inhibition of the expression of oncogenes, impairing the multiplication of tumor cells in vitro and promoting, under certain conditions, apoptosis, as well as the expression of tumor cell antigens important in either tumor growth and invasion or in the host immune surveillance (including MHC class I antigens). It is likely that these direct effects, mostly observed in cell culture models, can play a role also in vivo, in the context of an IFN-induced antitumor response, even though their importance and their possible relationships with the host mediated responses to IFN therapy remains unclear. The importance of the host-mediated antitumor effect was originally demonstrated in early studies in mice transplanted with syngeneic tumor cells showing that exogenous IFN-I was highly effective in inducing a potent antitumor response in animals bearing highly characterized tumor cells isolated for the in vitro resistance to IFN [24,25]. The role of specific host-mediated immune mechanisms was also demonstrated in an ensemble of studies performed in mice transplanted with genetically modified tumor cells expressing IFN-α [24]. These studies, in fact, revealed that the production of this cytokine at the tumor site resulted in a loss of tumorigenic and metastatic phenotype in different tumor models, which was mediated by multiple host immune antitumor mechanisms triggered by the cytokine released at the tumor site [24].

There is plenty of evidence in mouse models suggesting that also endogenous IFN-I can play an important role in tumor development and progression. The first evidence on the role of endogenous IFN-I in the restriction of tumor growth stems back to an early study, showing that injection of mice with antibodies to IFN-I enhanced the growth of several transplanted tumors [26]. In subsequent studies, the role of endogenous IFN-I in cancer immune surveillance and in general in the innate recognition of tumors by several immune cells has been unraveled [27], mostly by using models of IFNAR1 knock-out mice [28]. These studies demonstrated that endogenous IFN-I, either expressed at basal levels under physiological conditions or induced in the context of tumor growth, acted by suppressing tumor development and progression.

Notably, emerging data are also supporting the novel concept that IFN-I can act by suppressing the growth of cancer stem cells (CSC). In particular, by using Her2/Neu transgenic mice carrying a non-functional mutation in IFNAR1, it was recently demonstrated that an impaired IFN-I signaling results in increased amount of breast CSC, associated with enlarged vessels, in the absence of immune cell infiltrates [29]. Of note, IFNAR-neuT tumors specifically exhibited deregulation of genes having adverse prognostic value in breast cancer patients, including breast CSC marker aldehyde dehydrogenase-1A1 (ALDH1A1). In vitro exposure of neuT^+^ mammospheres and cell lines to anti-IFN-I antibodies resulted in increased frequency of ALDH^+^ cells, suggesting that IFN-I control stemness in tumor cells. This finding is consistent with in vitro data on human CSC [30] and with recent results showing a critical role for a mir-199-LCOR-IFN-I axis in breast CSC biology in mouse models [31].

The importance of endogenous IFN-I in the restriction of tumor growth is also supported by data in humans suggesting a role of IFN-I in tumor development and progression. Notably, human tumors exhibit defects in IFN-I signaling [32], often associated with a poor prognosis [33], whereas tumors exhibiting infiltrating T cells frequently show an IFN-I signature correlating with the clinical response [24,34]. Of note, IFN-I production can also be induced in tumor cells, as a result of stimulation by specific types of danger signals, including cell products released in response to some chemotherapy agents [35]. In principle, one could thus hypothesize that the mechanisms of antitumor action of the endogenous IFN-I can be identical or very similar to those identified in tumor bearing animals treated with exogenous IFN-I. However, several data emphasize that the mechanisms can markedly depend on doses and timing of IFN exposure as well as on the type of tumor models.

Today, it is also becoming clear that a successful response to conventional therapies, such as radiotherapy (RT) and chemotherapy, relies on the endogenous production of IFN-I. Burnette and co-workers showed that endogenous IFN-I played a crucial role for tumor eradication following RT in a mouse melanoma model [36]. This concept was then supported by further studies in mouse models reporting that RT-induced IFN-I increased the release of chemokines playing a role in the recruitment of lymphocytes at tumor site and enhanced the generation of CD8^+^ tumor specific effector cells [37]. Of interest, Sistigu and colleagues reported that anthracyclines can stimulate TLR3 in cancer cells by prompting an IFN-I signaling pathway [35]. Notably, an IFN-I-related signature was shown to predict clinical responses to anthracycline-based chemotherapy in patients with breast carcinoma characterized by poor prognosis. This study [35] also showed that the expression of the IFN-I-induced MX1 gene was upregulated by anthracyclines and strongly correlated with increased survival in breast cancer patients treated with anthracycline-based chemotherapy [35].

In view of the multiple biologic activities by which IFN-I can inhibit tumor growth, including induction of T cell immunity, it is not surprising that the clinical use of these cytokines (especially IFN-α) met a major success in patients with solid tumors expressing tumor associated antigens like melanoma and renal cancer, where the importance of an antitumor T cell response is currently envisaged. In other tumor models, such as chronic myeloid leukemia and breast cancer, the antitumor response can be, at least in part, mediated by direct effects on tumor cells and, in particular, on CSC. In contrast, a large majority of human malignancies have shown poor or no response to the clinical use of these cytokines, as a possible consequence of the resistance to direct effects on tumor cells as well as of tumor intrinsic immunosuppressive properties. Of interest, solid tumors showing high levels of tumor infiltrating lymphocytes (TIL) have been defined as “hot tumors” and are generally detected in some cancers with a high mutational load. “Hot tumors” tend to exhibit an “IFN-I signature”, which is often associated with a better prognosis or response to therapies [37,38]. Of note, the activation of the IFN-I system has not only been implicated in TIL accumulation and in promoting a T cell-inflamed tumor, but also in negative regulation of immune suppressor cells. While “hot tumors” are generally thought to be a therapy-responding tumor, a non cell-inflamed tumor (“cold tumor”) is unlikely to respond to checkpoint inhibitors [6,23]. Together, CD8^+^ T cell content and IFN-signature are both considered as important components in predicting a hot tumor microenvironment (TME) as well as response to immunotherapy [38]. Likewise, some studies indicate that the response to checkpoint inhibitors (CPI) are mediated by endogenous IFN-I [39,40]. Although the role of IFN-I in cancer has generally been reported as beneficial, necessary to both promote T cell responses and prevent metastases, reports have been published suggesting that, mostly in viral infections but also in cancer, sustained IFN-I signaling can have a negative role by promoting a tolerogenic circuit leading to immunosuppression and reduced efficacy of immune checkpoint blockade-based therapies [41,42]. Overall, these data highlight the complexity of the IFN-I system in promoting and inhibiting multiple environmental and cellular functions to modulate immunity in infections and cancer.

## 3. The Link between Obesity and Cancer

The escalating global epidemic of overweight and obesity is a major public health and economic problem, as excess body weight is associated with a consistent number of deaths and disability worldwide.

Overweight and obesity are significant risk factors for a number of chronic disorders, including cardiovascular diseases, hypertension, T2D, non-alcoholic fatty liver disease and several types of cancer. Obese subjects exhibit a higher risk of developing cancer, metastasis as well as decreased disease-free and overall survival [20,21]. There is currently strong evidence for a correlation between obesity and at least 13 different types of cancer [43].

Obesity is a heterogeneous chronic disease characterized by the accumulation of abnormal or excess adipose tissue (AT), which consists of adipocytes or fat cells, as well as immune cells, stromal cells, blood vessels, and neurons, that may impair health [WHO. Obesity and overweight. http://www.who.int/mediacentre/factsheets/fs311/en/ (accessed 11 August 2016)]. The link between obesity and cancer risk and mortality was firmly established by Calle and coworkers in a seminal study published almost 20 years ago [44]. Since then, many other epidemiologic reports highlighted a correlation between obesity and cancer (Diet, Nutrition, Physical activity and Cancer: A Global Perspective–Continuous Update Project Expert Report 2018—available at dietandcancerreport.org). Although several scenarios have been envisaged to explain this link, the mechanisms behind this connection remain to be fully elucidated. Among those proposed to link obesity and cancer are the effects mediated by insulin resistance, the up-regulation of sex hormones and of Programmed Cell Death Protein (PD)-1 protein, dysregulation of adipokine secretion [45]. Interestingly, obesity-induced alterations of the immune system can contribute to cancer establishment and progression as well as influence patient response to therapy. A main feature of this pathological condition is a chronic low-grade inflammation state, either local within the AT or systemic, which affects at various levels innate and adaptive immune responses [46]. Inflammation has been longer considered a key aspect in the pathogenesis of cancer [47] and compelling evidence has been achieved that an inflammatory state or in turn attenuation of inflammation may favor or prevent cancer development, respectively. Altered immune functions, both systemic and within the AT, have been consistently reported in obese individuals [48,49,50] and are thought to represent a main contributor to the obesity-associated greater risk for chronic diseases, including cancer, as well as to the increased prevalence and severity of common infections [51]. The immune dysfunctions observed under obesity rely on the excess AT and adipocyte hypertrophy that promote proinflammatory cytokines release, both locally and systemically [52], and excessive recruitment and infiltration of immune cells, especially macrophages [53,54]. Obese individuals exhibit lowered concentration of the anti-inflammatory adipokine adiponectin, and higher levels of the pro-inflammatory adipokine leptin together with other pro-inflammatory cytokines such as TNF-α, IL-6, CCL2, and IL-1β in AT.

More recently, a number of studies unraveled the capacity of obesity to affect the immunologic processes involved in host immune defense including alterations of lymphoid tissues, of immune cell development, differentiation and function, as well as of the coordinated action of innate and adaptive responses [46,55,56,57]. The integrity of immune tissue architecture is fundamental for the proper leukocyte generation and maturation. In the case of obesity, this architecture is altered by fat deposition in the tissues of the immune system, including the bone marrow and thymus, thus altering leukocyte distribution, lymphocyte activity and overall immune defense [55,56,58,59,60,61]. Furthermore, obesity and insulin resistance are associated with reduced thymic output in humans [56], decreased lymph node size and T lymphocyte numbers [62], and adversely affects the dynamics of secondary lymphoid tissues overall reducing the repertoire of circulating T cells, thus limiting the range of pathogenic antigens to which they can respond [55,60].

Another important element contributing to the disruption of the immune function in obesity is represented by the immunomodulatory effects resulting from the interaction of leukocytes with systemic markers of insulin resistance, chronic inflammation, and metabolic syndrome. In this regard, it is recognized that insulin resistance affects the processes involved in the resolution of inflammation since insulin plays a main role in inducing the differentiation of T helper cell 2 (Th2) endowed with anti-inflammatory properties [63], while hyperactivation of the immune response due to exaggerated uptake of glucose has been associate with cancer and autoimmunity [64].

Alterations in the distribution of leukocyte subsets, their inflammatory phenotypes, as well as of white blood cell (WBC) have been observed in obesity [55] Moreover, WBC counts, a parameter used to measure inflammation and leukocyte activation, are reduced by weight loss [65]. Of note, the frequencies of different T cell subsets as well as their blastogenic response have been reported to be altered in obese patients. In particular, a greater frequency of CD4^+^ T cells parallels a reduced frequency of CD8^+^ T cells [66]. Likewise, stimulation of T cells, CD3^+^, CD4^+^, CD8^+^, CD4^+^CD45RO^+^, and TCR αβ T cells with phytohemagglutinin or concanavalin A results in a weaker proliferative response in obese subjects. Furthermore, these subjects exhibit a negative correlation between the body mass index (BMI) and the level of TCR αβ as well as increased blood levels of TNF-α and soluble TNF-α receptors with respect to non-obese subjects, suggesting that dysregulated production of this cytokine may be at least in part responsible for the observed T cell dysfunction [67]. Likewise, peripheral blood mononuclear cells (PBMC) collected from obese subjects show a higher activation of NF-kB and NF-kB target genes (i.e, migration inhibition factor, IL-6, TNF-α, and matrix metalloproteinase 9) as compared to lean individuals [57].

Recent studies have also shown that the phenotype and function of NK cells are impaired under obesity. In particular, studies carried out in animal and human models reported that obesity is associated with alterations in the distribution, phenotype, cytotoxic activity, cytokine profile secretion, and signaling cascades of NK cell subsets [68], suggesting that these profound changes of NK cell biology can be of relevance in determining not only an increased cancer risk but also the severe cancer outcome in obese individuals [69]. It is noteworthy that obesity is associated with a reduced blood frequency of activated T regulatory (T_reg_) cells that parallels a concomitant enrichment of OX40-expressing T_reg_ cells in visceral AT (VAT), and directly correlates with BMI. In addition, obese individuals show a significant reduction of the Vγ9Vδ2/γδ T cell ratio at the systemic level [50] highlighting the role of obesity in the impairment of cell populations playing important functions in immune surveillance against tumors.

## 4. The Interplay between Type I Interferon and Obesity

A collection of data from clinical cases in humans and studies in mouse models highlight a rather complex role of the IFN-I system in obesity, mainly pointing to defects of IFN production/signaling as one possible explanation of the well-described enhanced susceptibility of obese subjects to infectious diseases and in the progression to obesity itself.

Obesity-associated immune dysfunctions are thought to represent important players in the increased susceptibility to viral infections, co-infections, and opportunistic infections involving several organisms such as Mycobacterium tuberculosis, Coxsackie virus, Helicobacter pylori, influenza and more recently SARS-CoV-2 viruses, observed in obesity [70,71,72]. In this regard, it has been reported that obese individuals exhibit a higher susceptibility and mortality during seasonal and epidemic flu infections. Obese individuals show delayed and blunted antiviral responses to influenza virus infection, characterized by more severe lung inflammation and damage from viral pneumonia, and they experience poor recovery from the disease [72]. Studies in human cohorts and animal models have highlighted a prolonged viral shed in the obese host, as well as a microenvironment that permits the emergence of virulent minor variants. The reduced efficacy of antivirals and vaccines in obese subjects suggests that obesity may also play a role in altering the viral life cycle, thus complementing the already weakened immune response and leading to severe disease [72]. Multiple immune responses including IFN-I production and IFN-signaling of IFN stimulated genes (ISG) are impaired in the respiratory epithelial cells and macrophages of obese individuals, together with an increased production of inflammatory cytokines and M1 polarization of lung macrophages. Thus, the deficiency of IFN production and signaling in obese patients might be among the risk factors for severe outcomes in pandemic influenza infection [72,73,74].In this regard, an aberrant IFN-I response during some viral infections and after TLR stimulation has been reported to impair its antiviral efficacy. PBMC from obese subjects exhibit a diminished ability to produce IFN-I as well as pro-inflammatory cytokines after TLR stimulation and influenza A/H1N1 infection. This aberrant response directly correlates with a high expression of the suppressor of cytokine signaling-3 (SOCS3) but not of SOCS1 in obese subjects [75,76], suggesting that SOCS3 plays a role in inducing this diminished response. SOCS3 is a key regulator of IFN-I as well as of leptin and pro-inflammatory cytokines, which are elevated in obesity [77]. Thus, SOCS3 upregulation and altered systemic leptin levels could be responsible for the reduced IFN-I response as well as for other immune dysfunctions relevant to T and B cells in people with obesity. In keeping with this hypothesis, PBMC from obese volunteers, silenced for SOCS3, show increased IFN-α expression and production. Conversely, no increase in the IFN-α response has been observed in the non-obese volunteers, thus confirming the key role of SOCS3 in inhibiting IFN-I after TLR stimulation in obesity [78]. In addition to the described obesity-associated impairment of IFN-I response leading to decreased antiviral efficacy, the existing literature highlights a reciprocal regulation between IFN-I and metabolic processes [12,13].

In this regard, it has been described that IFN-I/IFNAR axis is involved in obesity-associated AT inflammation with either detrimental or beneficial effects (Table 1).

Among the detrimental effects, Ghosh and colleagues reported that in obese individuals, adipose-derived chemerin can recruit plasmacytoid DC (pDC) from the circulation into the VAT, thus linking the hyperadiposity-driven functional phenotype of adipocytes to recruitment of innate immune cells [79]. IFN-I produced by pDC in response to AT–derived HMGB1 and extracellular self-DNA molecules in turn fuels metaflammation by driving proinflammatory polarization of macrophages in VAT and contributes to systemic insulin resistance. In keeping with these results, depletion of pDC and abrogation of IFN-I signaling prevent diet-induced obesity and T2D [85]. Furthermore, obesity-induced expression of IFN-I in the liver has been reported to drive the accumulation and activation of intrahepatic CD8^+^ T cells to promote metabolic syndrome [90]. Likewise, Chan and coworkers showed that IFN-I receptor engagement by IFN-I produced by primary adipocytes, either of mouse or human origin, modifies the adipocyte-intrinsic metabolome to shape inflammatory vigor and immune responses [80]. In this regard, it has been shown that deficiency of IRF7, a master regulator of IFN-I induction, prevents high fat diet (HFD)-induced obesity and insulin resistance pointing to the involvement of this factor in diet-induced alterations in energy metabolism and insulin sensitivity [81]. More recently, the involvement of IRF7 in the pathogenesis of obesity was related to its capacity to regulate CCL2 expression [82]. On the same line, evidence has also been achieved on the involvement of IRF5 in obesity-associated events such as fat accumulation, insulin resistance and polarization of macrophages toward an M1 proinflammatory phenotype [91]. Moreover, a positive correlation has been reported between the increased expression level of IRF5 in the AT of overweight/obese individuals and that of inflammatory markers [83].

Conversely, it was shown that IFN-β1 over-expression as well as IFN-α-2b administration in mouse models of HFD-induced obesity prevent weight gain and suppress immune cell infiltration into AT, attenuate adipose inflammation and limit AT expansion [87], or increase fatty acid oxidation and reduce cholesterol levels [88], respectively. Likewise, in a diet-induced obesity mouse model, the administration of IFN–tau, a member of IFN-I family, results in increased insulin sensitivity, decreased expression of pro-inflammatory cytokines and expansion of anti-inflammatory M2 macrophages [89]. Furthermore, antisense oligonucleotide blocking of T39, a scaffolding protein promoting the ubiquitination and degradation of liver X receptor, produces an off-target IFN-I response that protects against diet-induced obesity [84]. Likewise, another study reported that adipose IFN-I signaling protects against metabolic dysfunction, obesity and hepatic disease in mice exposed to a high-fat or methionine-choline-deficient diet [86]. Interestingly, weight loss and glycemic control induced by laparoscopic gastric banding in patients with severe obesity is associated with increased IFN-I responses in hepatic and adipose tissues.

## 5. Focusing on One Specific Cancer Model: Colorectal Cancer

While obesity is currently considered as a general risk factor for the large majority of cancer types, the evidence on the importance of the role of this pathological condition in cancer development and progression is particularly documented in some neoplastic diseases. Among them, colorectal cancer (CRC) is one of the solid tumors where this evidence is strongly demonstrated by an ensemble of studies in both animal models and humans. CRC is the third most commonly diagnosed cancer and the second cause of cancer-related mortality worldwide, thus representing a considerable health issue (Globocan cancer statistics 2018). Several factors including genetic, environmental and lifestyle factors, have been reported to play a key role in CRC etiology. Among these, the interplay between body weight, dietary patterns and CRC risk is one of the strongest for any type of cancer with important implications for prevention strategies. There is currently agreement on the fact that staying physically active, maintaining a healthy body weight and eating a healthy diet can prevent CRC development, thus highlighting that the CRC risk is highly modifiable through lifestyle (Diet, Nutrition, Physical activity and Cancer: A Global Perspective–Continuous Update Project Expert Report 2018–available at dietandcancerreport.org). Noteworthy is that the incidence/mortality of CRC in specific geographical areas may reflect economic and societal changes, including the adoption of more Western lifestyles and the growing obesity rates. Excess adiposity is strongly linked to CRC as it represents an important indicator of disease outcome. The increased CRC risk (i.e., 1.5–3.5-fold) observed in obese subjects as compared to lean individuals is mostly related to abdominal rather than overall obesity, that is considered to be more predictive of CRC risk [92]. Inflammation is an important player in CRC pathogenesis [93] and the low-grade chronic inflammation characterizing obesity has been long considered an important determinant for CRC risk. The link between inflammation and cancer is further supported by the observation that anti-inflammatory drugs lower CRC risk and retard intestinal tumors in ulcerative colitis patients [94]. In this regard, compelling evidence has been achieved on the important role of diet in CRC initiation and progression that relies on its potential to modulate inflammation, both locally in the AT and systemically by regulating several immune and inflammatory pathways. Furthermore, the composition of the intestinal microbiota is strongly influenced by diet. Some studies reported a reduction in diversity and richness of microbiota—termed dysbiosis—in obese individuals that correlates with low-grade inflammation, increased body weight and fat mass, and T2D [95,96]. Likewise, intestinal commensal flora can directly promote carcinogenesis by sustaining local mucosal inflammation or favoring systemic metabolic/immune dysregulation or indirectly, through the modulation of the anti-tumor response. CRC patients at different stages show alterations of fecal and mucosal microbiota characterized by a marked reduction of bacterial diversity [97,98]. The ensemble of these observations points to diet as an important factor in controlling the composition of the intestinal microbiota, that in turn not only maintains the immune homeostasis but can also contribute to colorectal carcinogenesis [99,100,101]. IFN-I are produced in normal gut mucosa as well as in the TME and the IFN response is tuned by the commensal microbial community [102]. Constitutive IFN-I contribute to intestinal barrier function, drive IgA against commensal bacteria, and regulate intestinal macrophage function. Through these mechanisms, constitutive IFN-I signaling may be essential for maintaining intestinal immune homeostasis by enhancing innate responses to bacteria, increasing intestinal barrier functions, and producing factors that prevent intestinal dysbiosis [103]. Recent studies suggest that expression and secretion of IFN-I in the TME is a key player in the antitumor cascade and that the measurement of IFN-I signals and signatures (i.e., regulators and targets) of this pathway could serve as a prognostic biomarker [104]. Tumor-induced suppression of IFN-I signaling in the TME impairs anti-CRC immunity and correlates with poor disease outcome [33]. Likewise, the expression of IFN-I stimulated genes (IRF1 and 2, IFITM1) has a prognostic value and has been associated with CRC risk, metastasis and patient survival [105,106,107,108]. Interestingly, IFN-I signaling in cancer and immune cells is a major mediator of the antitumor response induced by chemotherapy/RT, and IFN-I inducers have promise in increasing patients response to CPI [104]. Lastly, IFNAR1 is a predictor for overall survival and its mRNA expression is correlated to IRF7, but not TLR9, in CRC [109]. Despite growing evidence on the role of IFN-I in the antitumor response as well as on its prognostic potential in CRC, IFN-I-dictated molecular and immune signatures are still poorly characterized in this tumor and their prognostic role remains to be established in prospective and multicentric studies. Currently, only few of the studied CRC biomarkers have been transformed into clinically validated diagnostic/prognostic tools. The development of novel biomarkers defining the molecular mechanisms governing immune reactivity and predicting their relationship to treatment would offer new perspectives for more personalized and clinically effective treatments. In this scenario, tumor inherent IFN-I are capable of promoting an immune reactive TME and have promise as prognostic biomarkers in oncology. Furthermore, due to the well-recognized role of gut microbial composition and lifestyle in CRC risk and disease outcome, understanding their relationship with IFN-I-dictated signatures in obese and control individuals would advance the comprehension of the mechanisms underlying CRC heterogeneity in pathogenesis and therapy response.

## 6. Perspectives and Future Directions

A great research interest is currently focused on the role of pleiotropic cytokines such as IFN-I in the control of both tumor growth and obesity, especially in view of the well-documented relationship bemouse and human modelstween cancer and metabolic dysfunctions leading to excessive body weight and abnormal fatty tissue accumulation. Today, many data are available on the effect of endogenous IFN-I in inhibiting tumor development and growth in both mouse models and humans highlighting their crucial role not only on tumor growth but also on the response to different antitumor therapies. As a matter of fact, activation of IFN-induced genes in the TME represents an emerging biomarker of the antitumor response in patients. In contrast, hyperactivation of IFN-I system due to chronic exposure to the cytokines within the TME as well as in peripheral blood cells and some tissues may play a detrimental role. Notably, the role of IFN-I in obesity appears to be even more complex. As an example, IFN-I can prevent HFD-induced obesity in mouse models, while IFN-I produced by pDC recruited in the AT can shape the response towards inflammatory events leading to metabolic dysfunctions and obesity itself. However, it should be noted that while the role of IFN-I in anti-tumor response is well-documented by several human studies, the involvement of IFN-I in weight control and metabolic regulation has begun to be documented only in more recent years, mainly in mouse models. Nevertheless, growing evidence suggests that IFN-I plays a role in obesity-related inflammatory events as well as in metabolic and body weight regulation. Figure 1 schematically depicts the complexity and the so-called “Janus face” of the IFN-I system with respect to obesity and cancer, which may result in different and even opposite effects under different physiological and pathological conditions, likely depending on the site, IFN-I subtype and quantity of local versus systemic cytokine production, thus highlighting the importance of the fine regulation of IFN-I signaling in different cells and tissues.

A schematic representation of the effects of IFN-I system dysregulation, likely depending on the site of occurrence and extent of expression, which may result in different and even opposite effects in pathological conditions such as cancer and obesity.

It is of interest to note that obese subjects are more susceptible to viral infections, often associated with a lower IFN production, and have a lower response to standard antitumor therapies, including chemotherapy [110]. In view of the well-known role of IFN-I in protecting from viral infection and in mediating the antitumor response to several therapeutic regimens, all this is suggestive of a lower level of endogenous IFN response in obese subjects with respect to healthy individuals. It is worth noting, however, that obese cancer patients are more responsive to the treatment with CPI than normal patients [111]. This is an intriguing observation, which may be in part explained by both the PD-L1 up-regulation on myeloid-derived suppressor cells and enhancement of PD-1 expression on T cells in the TME as a result of inflammatory response primarily occurring in the AT. Notably, however, it has recently been suggested that hyperglycemia can result in increased cardiotoxicity in response to CPI and enhanced production of inflammatory cytokines including IFN-γ production [112]. Furthermore, studies in mouse models and humans point to IFN-I as a key modulator of the inflammatory response observed at early phases of type 1 diabetes development [113]. Although both type 1 and type 2 diabetes are recognized risk factors for diverse tumors [114,115], there is nowadays no clear evidence for a direct involvement of IFN-I in these pathogenic processes. All this reveals the complexity of the interplay between obesity, cancer and response to different antitumor therapies. Further and well controlled studies are needed to better understand the impact of factors such as diet, microbioma, genetic make-up and metabolic microenvironment at different tissue levels, including TME and AT, in affecting IFN-I production and response. Evidence suggests that the fine tuning of IFN-I response is regulated by energy, lipid and amino acid metabolism [12]. Loss of this fine-tuning can result in sustained IFN signaling, immunosuppression and tissue damage, which has been implicated in pathogenesis of chronic viral infections, autoimmune diseases and cancer [18]. In this regard, it is worth of note that incorrect eating behaviors play an important role in tumor development, pointing to diet as a key determinant for tumor prevention. However, no data are still available, to the best of our knowledge, on the influence of specific dietary habits or food components on the IFN-I system. Studies in this research area will be pivotal to fully understand the role of the IFN-I system as an additional player in the relationships between cancer and obesity.

While today IFN-I have been largely replaced by new drugs in the clinical practice in patients with cancer and viral or autoimmune diseases, the most recent knowledge on the complex roles of these cytokines under different pathologic conditions reveals new and more selective strategies for their clinical use [23,116]. Novel and potentially more effective modalities of using these cytokines should take into account the IFN effects on cell metabolism at different tissue levels and under various physiological conditions, including hyperglycemia and obesity. Currently, a few drugs have been tested and eventually approved for the treatment of obesity and most of them have revealed a relevant toxicity [117]. Thus, future research challenges may also include the identification of how new treatment modalities selected for ensuring a fine regulation of the IFN-I system can prevent metabolic dysfunction and pathologic conditions, including obesity, potentially leading to cancer development and growth. The achievement of this goal will require a combination of preclinical studies in both in vitro and in vivo mouse models together with well-designed proof-of-concept clinical trials and strategic collaborative research efforts for clinical validation in patients.

## 7. Conclusions

Increasing evidence suggests today the relationship between cancer and metabolic dysfunctions including obesity. IFN-I play a well-recognized role in cancer development, progression and response to therapeutic treatments. Notably, these cytokines markedly affect the metabolism of both tumor and host cells and can either promote or inhibit obesity, depending on the site and modalities of production and the exposure time in different tissues. Further in vitro and in vivo studies are needed to dissect the mechanisms by which the IFN-I system can affect tumor growth and metabolic functions under both physiological and pathologic conditions including obesity. This knowledge could be translated into clinical applications for controlling both tumor growth and obesity. 

## Figures and Tables

**Figure 1 cancers-13-00196-f001:**
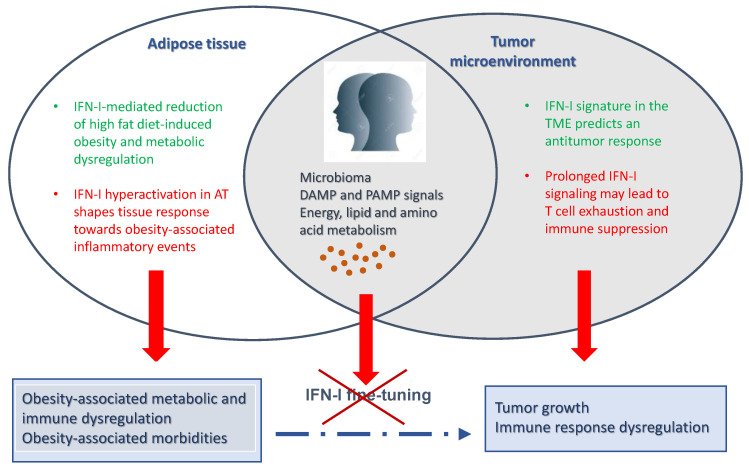
The Janus face of IFN-I signaling in cancer and obesity.

**Table 1 cancers-13-00196-t001:** The dual role of IFN-I in obesity.

Cell Type/Tissue	Study Model	Observed Effect	References
Visceral AT	Obese subjects	↑ IFN-I expression by VAT-recruited pDC ↑ ISG expression in obese VAT	[79]
Mouse/human primary adipocytes	HFD WT and IFNAR^−/−^ mice; Obese subjects undergoing bariatric surgery	↑ adipocyte IFN-I signature ↑ LPS-induced IL-6 production in IFNβ-primed adipocytes from HFD mice plasmacytoid= obesity degree ^1^ but altered white AT distribution in HFD IFNAR^−/−^ and WT mice ↓ IFNAR signaling-dependent proinflammatory cytokine-producing macrophages	[80]
Visceral AT (epididymal, mesenteric and perirenal)	HFD WT and IRF7^−/−^ mice	↑ IRF7 expression in obese mice AT ↓ weight gain and adiposity ↑ glucose and lipid homeostasis and insulin sensitivity on HFD in IRF7 ^−/−^ mice ↓ diet-induced hepatic steatosis	[81]
Epididymal AT and 3T3-L1 pre-adipocytes	WT and IRF7^−/−^ preadipocytes and HFD mice	IRF7 regulates AT CCL2 expression HFD ↑ CCL2 more in WT than in IRF7^−/−^ mice	[82]
Subcutaneous AT	Obese subjects	↑ AT IRF-5 expression in obesity correlates with TNF-α and CCL5 levels, BMI, body fat percentage, age, HbA1c, systemic immuno-metabolic markers ↑ AT CXCL8 expression in obese individuals is associated with IRF5 expression IRF5 expression is associated with inflammatory/immune marker signature in AT	[83]
AT macrophages	HFD WT and IFNAR^−/−^ mice	↓ HFD-induced obesity upon promotion of local IFN-I response in AT macrophages by antisense oligonucleotides	[84]
Liver, spleen and adipose tissues	HFD WT and IFNAR^−/−^ mice	Abrogation of IFN signaling and pDC depletion ↓ HFD-induced obesity and T2D	[85]
Mouse adipocytes Human subcutaneous AT	HFD WT and IFNAR^−/−^ mice Obese subjects	↑ IFN-I-regulated gene expression by HFD in WT mice liver protects against metabolic dysregulation Bariatric surgery-induced weight loss restores IFN-I responses and reduces metabolic dysregulation in severe obesity	[86]
Liver and adipose tissues	IFNβ1 overexpression, IFN-α-2b or IFN-tau administration in HFD mouse models	↓ HFD-induced adipose hypertrophy, inflammation and weight gain Altered gene expression in AT toward a thermogenic phenotype Restoration of insulin sensitivity and improvement of glucose homeostasis but no rescue of HFD-induced fatty liver ↑ fatty acid oxidation and M2 macrophages ↓ cholesterol levels, pro-inflammatory ciyokines	[87,88,89]
PBMC	Obese subjects	↓ IFN-α2 and IFN-α6 production in response to TLR engagement in obese subjects ↑ SOCS3 expression in obese subjects	[75]
PBMC	Influenza virus infected obese subjects	↓ IFN-β production in response to TLR3 ligands in obese subjects = IFN-α production in response to TLR7 ligands in obese and lean subjects	[76]

^1^ i.e., body weight energy expenditure, food intake, systemic cholesterol, total body adiposity, ↑enhancement, ↓decrease, total BAT Ucp-1 expression and white AT morphology. Abbreviations: adipose tissue (AT); high fat diet (HFD); plasmacytoid dendritic cells (pDC); IFN-stimulated gens (ISG); IFN-I receptor (IFNAR1); wild-type (WT); peripheral blood mononuclear cells (PBMC); suppressor of cytokine signaling-3 (SOCS3); brown adipose tissue (BAT); anti-uncoupling protein-1 (Ucp-1); glycated hemoglobin (HbA1c); type 2 diabetes (T2D).

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
