# Peer review of "Type I Interferons as Joint Regulators of Tumor Growth and Obesity"

_cancers, 2021, doi:10.3390/cancers13020196_

Round 1
Reviewer 1 Report
The review titled "Type I interferons as joint regulators of tumor growth and obesity" is a well written description of the role of cancer and adypocyte microenvironment in cell metabolism regulation. The review is well written, references are sufficient and updated. However, i suggest authors to include detailed information on the role of hyperglycemia in cytokine profile in cancer cells/tissue or adipose cells (Int J Mol Sci. 2020 Oct 21;21(20):7802); moreover, i suggest to describe the role of interferons and other cytokines in non cancer-cancer patients with metabolic syndrome and describe the putative therapeutic strategies available to date.
Author Response
The review titled "Type I interferons as joint regulators of tumor growth and obesity" is a well written description of the role of cancer and adipocyte microenvironment in cell metabolism regulation. The review is well written, references are sufficient and updated. However, I suggest authors to include detailed information on the role of hyperglycemia in cytokine profile in cancer cells/tissue or adipose cells (Int J Mol Sci. 2020 Oct 21;21(20):7802); moreover, I suggest to describe the role of interferons and other cytokines in non cancer-cancer patients with metabolic syndrome and describe the putative therapeutic strategies available to date.
As a follow up to the Reviewer’s comment, information on the role of hyperglycemia in cancer and on the role of IFN-I in metabolic syndrome and available strategies has been added in Section 4. “The interplay between type I interferon and obesity” and Section 6. “Perspectives and future directions” together with related references.
Reviewer 2 Report
The topic of this paper is of great importance and is well written. However, the following proposed revision may significantly improve the overall quality of the publication.
1) Abstract: “IFN-I exhibit different and even opposite effects on obesity”. Elaborate the sentence. “Similarly localized IFN-I expression may result in inhibition of diet-induced obesity”. Localized to which tissues? Avoid the use of the word ‘may’.
2) The review is not distinguishing between obesity and types of cancers or the reciprocal relations between them. Briefly, it should be described somewhere in the section “The link between obesity and cancer”.
3) Add a few lines about the correlation between IFN-I, diabetes, and cancer.
4) Describe the roles of IFN-I in a diverse type of cancers
5) Add a figure showing mechanisms or effector molecules by which IFN-I regulates tumor growth and Obesity
6)What is the effect of anti-obesity drugs on the level of IFN-I. Describe it. similarly, describe the effect of antitumor drugs on the IFN-I level. In vivo and clinical data
7) Add section “Perspective and future directions”
Author Response
Abstract: “IFN-I exhibit different and even opposite effects on obesity”. Elaborate the sentence. “Similarly localized IFN-I expression may result in inhibition of diet-induced obesity”. Localized to which tissues? Avoid the use of the word ‘may’.
The abstract has been revised accordingly to the Reviewer’s comment.
The review is not distinguishing between obesity and types of cancers or the reciprocal relations between them. Briefly, it should be described somewhere in the section “The link between obesity and cancer”.
This information has been added in Section 3. “The link between obesity and cancer” and further discussed in Section 5. “Focusing on one specific cancer model: colorectal cancer “.
Add a few lines about the correlation between IFN-I, diabetes, and cancer.
Accordingly to the Reviewer’s suggestion, the correlation between IFN-I, diabetes and cancer is now briefly discussed in the Section 6. “Perspectives and future directions”
Describe the roles of IFN-I in a diverse type of cancers
This information has been included in Section 2. “Type I interferons and cancer”
Add a figure showing mechanisms or effector molecules by which IFN-I regulates tumor growth and Obesity
In Figure 1 we have schematically depicted the complexity of the IFN-I system with respect to obesity and cancer, which may result in different and even opposite effects under various physiological and pathological conditions, in order to highlight the importance of the fine regulation of IFN-I signaling in different cells and tissues. However, we did not consider necessary to graphically represent the mechanisms/effector molecules underlying the IFN-I-mediated regulation of cancer growth as they are largely described in the text and have already been the subject of dedicated reviews. With respect to obesity, the role of IFN-I, either promoting or protecting, is still matter of debate and univocal data on mechanisms/effector cells are currently lacking.
What is the effect of anti-obesity drugs on the level of IFN-I. Describe it. similarly, describe the effect of antitumor drugs on the IFN-I level. In vivo and clinical data
To the best of our knowledge, there are no data on the effects of anti-obesity drugs on IFN-I level. The importance of assessing such parameter in future studies has now been highlighted in the Section “Perspectives and future directions”. Conversely, the capacity of anti-tumor drugs to modulate the IFN-I system is well known and described to a large extent in Section 2. “Type I interferons and cancer”.
Add section “Perspectives and future directions”
As a follow up to the Reviewer’s suggestion, we have revised and implemented the Section “Conclusions” by adding sentences on perspectives and future directions.